# Pregnant Women Living with Obesity: A Cross-Sectional Observational Study of Dietary Quality and Pregnancy Outcomes

**DOI:** 10.3390/nu13051652

**Published:** 2021-05-13

**Authors:** Margaret Charnley, Lisa Newson, Andrew Weeks, Julie Abayomi

**Affiliations:** 1School of Applied Health and Social Care & Social Work, Faculty of Health, Social Care & Medicine, Edge Hill University, Ormskirk L39 4QP, UK; Charnlem@edgehill.ac.uk; 2School of Psychology, Faculty of Health, Liverpool John Moores University, Liverpool L3 3AF, UK; L.M.Newson@ljmu.ac.uk; 3Sanyu Research Unit, Department of Women’s and Children’s Health, University of Liverpool, Liverpool L7 8TX, UK; aweeks@liverpool.ac.uk

**Keywords:** pregnancy, obesity, nutrition, diet, BMI, malnutrition, micronutrients, macronutrients, clinical outcomes

## Abstract

Good maternal nutrition is key to optimal maternal and foetal health. A poor-quality diet is often associated with obesity, and the prevalence and severity of maternal obesity has increased significantly in recent years. This study observed dietary intakes in pregnant women living with obesity and assessed the quality of their diet. In total, 140 women with a singleton pregnancy, aged > 18 years and BMI ≥ 35 kg/m^2^, were recruited from antenatal clinics, weighed and completed food diaries at 16-, 28- and 36-weeks’ gestation. Clinical data were recorded directly from the women’s medical records. Nutrient intake was determined using ‘Microdiet^TM^’, then compared to Dietary Reference Values (DRVs). Energy intakes were comparable with DRVs, but intakes of sugar and saturated fatty acids were significantly higher. Intake of fibre and several key micronutrients (Iron, Iodine, Folate and Vitamin D) were significantly low. Several adverse obstetric outcomes were higher than the general obstetric population. Women with obesity, often considered ‘over nourished’, may have diets deficient in essential micronutrients, often associated with poor obstetric outcomes. To address the intergenerational transmission of poor health via poor diets warrants a multi-disciplinary approach focusing away from ‘dieting’ onto positive messages, emphasising key nutrients required for good maternal and foetal health.

## 1. Introduction

In England, the majority of adults are overweight or obese [1], and within the pregnant population, less than half are recorded as having a healthy weight, with over a fifth of pregnant women living with obesity [2]. The World Health Organization (WHO) [3] define the degree of overweight or obesity in adults using the following classifications: Healthy weight—Body Mass Index (BMI) 18.5 kg/m^2^ to 24.9 kg/m^2^; Overweight—BMI 25 kg/m^2^ to 29.9 kg/m^2^; Obesity Class I—BMI 30 kg/m^2^ to 34.9 kg/m^2^; Obesity class II—BMI 35 kg/m^2^ to 39.9 kg/m^2^ and Obesity class III—BMI 40 kg/m^2^ or more [3] these classifications are also applied to clinical guidelines in England and Wales [4]. The Centre for Maternal and Child Enquiries (CMACE) and Royal College of Obstetricians and Gynaecologists (RCOG) jointly state that overweight and obesity during pregnancy significantly influence maternal mortality, with 49% of deaths occurring in women who are overweight (BMI ≥ 25 kg/m^2^), and 27% of deaths occur in women living with obesity (BMI ≥ 30 kg/m^2^) [5]. Obesity is often associated with poor quality diet. The World Health Organisation [6] acknowledges that good nutrition during early life is the most important factor in tackling both the double burden of disease and health inequalities worldwide. They assert that poor nutrition during early life, including pregnancy, can have detrimental, short-term and long-lasting effects. Maternal nutritional intake is an important determinant of Gestational Weight Gain (GWG), which influence maternal and child health outcomes.

In the UK, there are currently no official evidence-based guidelines for GWG during pregnancy, although the RCOG (2018) has acknowledged that there is “a lack of consensus” on optimal GWG, and therefore recommend that pregnant women living with obesity are encouraged to adopt a healthy diet rather than GWG targets, and receive advice from a dietitian [2]. However, in the absence of such guidance and limited access to dietetic input, Health Care Professionals (Midwives) often refer to international guidelines to inform their advice and monitoring of pregnant women [7]. Specifically, the USA Institute of Medicine guidelines [8] advises women to limit GWG according to pre-pregnancy BMI, as considered to be associated with optimal birth weight. The IOM recommends that pregnant women living with obesity (BMI ≥ 30 kg/m^2^) should limit their GWG to 11–20 lbs (5–9 kg) in total, and those classified as overweight (BMI ≥ 25 kg/m^2^) to 15–25lbs (7–11.5 kg) [9].

Despite levels of obesity in the UK, evidence reviewing dietary intake for women (aged 19–64 years old) in the UK National Diet and Nutrition Survey (NDNS) [10] reports overall sub-optimal diets with many women not consuming the recommended daily intake of fruit and vegetables, dietary fibre and omega-3-Fatty Acids; whilst intake of sugars and saturated fatty acids are too high. Moreover, with reference to specific micronutrients, such as Iron, intake is also lower than recommended. If these data are considered in the context of women within reproductive age, dietary intakes during pregnancy may remain sub-optimal or indeed decline further, given the increase in nutrient requirements of the mother and foetus during pregnancy. Deficiencies can negatively impact the health of the mother and the baby. Research examining the diet of pregnant women has increased in recent years and suggests issues in energy balance and nutrient intakes, with energy requirements being derived mainly from fat and protein with large contributions from saturated fat and sugar but with significantly lower intakes of dietary fibre, influencing maternal BMI and GWG but also increasing maternal metabolic stress and the risk for obesity in the offspring [11].

Further to this, intakes of micronutrients are known to influence pregnancy outcomes and neonatal health. Folate, found in green leafy vegetables and citrus fruits, is essential for cell division, protein synthesis and neurotransmitters in embryonic and foetal growth and development. Deficiencies are linked to neural tube defects and insufficient intakes to impaired neurological development [12]. To avoid deficiency, UK women are recommended to take folic acid supplements (400 µg per day); before conception until at least 12-weeks’ gestation [13]. Many other countries also recommend supplementation (dose ranges from 300–600 µg/day), with WHO recommending 600 µg/day [6]. Other countries, including the USA, have mandatory fortification of flour with folic acid to increase population intakes [14]. Iodine, derived from seafood, fortified foods and milk, is required for cognitive development. Deficiencies can lead to a range of outcomes from mild cognitive impairments to severe intellectual disability [15]. Iron deficiency anaemia is associated with low birth weight (LBW) or small for gestational age (SGA) infants, pre-term deliveries and abnormal psychomotor development and impaired cognitive function as well as increased risk of maternal infection [16]. Low maternal calcium status and low dietary intakes increase the risk for hypertensive disorders of pregnancy, and LBW and vitamin D, required for maintaining calcium homeostasis and optimising bone health, also plays an essential role in glucose metabolism, immune function, inflammation and regulation of gene transcription and expression [16]. In the UK, supplements of 10 µg per day of Vitamin D are recommended throughout pregnancy and lactation [13]. Globally, there is wide variation regarding recommended dose for vitamin D supplementation in pregnancy (ranging from 5–25 µg per day), with the WHO recommending 5 µg per day [6]. However, research evaluating the quality of diet for pregnant women living with obesity is limited. Notable studies assessing the quality of diet in the pregnant population are the Australian WATCH study [17], and LIMIT study [18], the US Project Viva [19], all of which assessed diet quality in pregnant women using food frequency questionnaires. Most studies are aimed at reducing risks, i.e., gestational diabetes, associated with GWG and obesity [20,21,22,23] via lifestyle interventions or reducing GWG [24,25,26].

There is a paucity of studies evaluating the quality of dietary intake in pregnant women living with obesity. Therefore, this study aimed to observe dietary intakes in a cohort of pregnant women living with obesity and assess the quality of their actual diet. It was hypothesised that dietary intakes of key nutrients may exceed or fail to meet dietary reference values (DRV).

## 2. Materials and Methods

### 2.1. Study Design

This was a cross-sectional observational study. This research and the write up of this study has applied STROBE guidelines (2009) to aid quality [27]. An NHS ethics application was approved for this study (IRAS-ref number 09/H1005/23).

### 2.2. Study Population

The study was part of a larger research project, the Fit for Birth Trial (FFB) where one of the outcomes was to explore associations of weight change during pregnancy in women living with obesity (BMI ≥ 30 kg/m^2^) with pregnancy outcomes. The sample size for the FFB trial initially recruited 824 women and collected mother’s weight during trimester one for 756 women at routine antenatal appointments. The results of the FFB trial are reported elsewhere [28]. This study opportunistically sampled participants through the FFB cohort, specifically recruiting those women referred into obstetric services for additional antenatal support as deemed a high-risk pregnancy (referral criteria BMI ≥ 35 kg/ m^2^) and thus an ideal target audience for dietary assessment. The sample was restricted to this identified cohort of 225 eligible women, who were all invited to extend their research participation into the nutritional data trial (FFB plus).

Recruitment took place through a large, inner-city maternity hospital and in satellite community clinics by the booking in midwives, who were fully briefed with regards to the study protocol. Pregnant women aged 18 years and over, with a recorded BMI ≥ 35 kg/m^2^, attending their initial antenatal booking-in appointment, between 12 and 14 weeks gestation, were asked to consent. Participants provided permission to collect data from medical records about pregnancy outcomes, mothers’ age, mothers’ weight, Body Mass Index (BMI) and birth weight. The participants consented to the nutritional research, agreeing to collect data regarding their daily dietary intakes. Exclusion criteria included women below the age of 18 years, those with multiple pregnancies, non-English speaking women and women with complex medical disorders, including those having undergone bariatric surgery. Women diagnosed with Gestational Diabetes Mellitus (GDM) following oral tolerance glucose testing at 28 weeks were referred to the Dietitian. However, no women were excluded from the study for the reasons stated. The recruitment cycle for the study took place over 12 months. Written consent was obtained, and the women advised that they could leave the study at any point without affecting routine antenatal care.

### 2.3. Dietary Measurements

Participants completed 3-day estimated food diaries at three different time points: 16-weeks, 28-weeks and 36-weeks’ gestation. Three-day food diaries were selected in preference to alternative food frequency questionnaires [29]. It has been argued that for longitudinal studies, estimated food diaries include more food details and are better able to demonstrate a relationship between food choices and nutrients compared to questionnaire methods [30]. Estimated food intake diaries were encouraged to reduce participant burden and increase data return. Maternal weight change data were collected via community midwives during routine antenatal appointments. Data relating to age, weight, BMI, and birth weight were collated from the women’s medical health records. Maternal weight change was recorded at approximately 36-weeks’ gestation, and birth weight data recorded following delivery of the baby.

### 2.4. Procedure

The diaries were posted out to participants 10 days before a research appointment that took place in a hospital setting. The instructions for these food diaries asked the women to record all food and drinks consumed over three days (including one weekend day and two weekdays) in as much detail as possible, specifying cooking methods and approximate portions.

Further detail regarding food items and portion sizes were verified with the participants during the follow-up appointment with the lead researcher (a registered nutritionist trained in food analysis, first author). During this appointment, the researcher used tools, such as images of household measures (e.g., spoons, cups; a photographic atlas of food portion sizes [31] and nutritional labelling on food packaging (looked up via supermarket websites) to aid discussion and clarify diary entries. For example, terms such as ‘butter’ were commonly used to describe all types of ‘spread’. The appointment allowed discussion so that terminology and amounts of food consumed could be confirmed, and this process aided the validation and reliability of dietary intakes recorded.

### 2.5. Data Analysis

Data regarding individual and composite foods recorded in the 3-day diaries were then individually coded by the lead researcher. Recipes for home-cooked composite meals were assessed, and the weight of servings approximated to allow for wastage (for example, including peel, seeds, stalks) during preparation. The Food Standard Agency, Food Portion Sizes guide [32] was used to estimate weight loss/gain differences between raw/uncooked and cooked ingredients before coding. Coded data was transferred into a food composition software package Microdiet™ (Downlee V1.1) [33] based in the UK on McCance and Widdowson Composition of Foods 6th edition [34] and used by universities, dietitians and nutritionists. The accuracy of data coding was checked by the senior and supervising Dietitian (last author). Food content was assessed for total daily energy intake in kcals for each participant. Macronutrients that contribute to total energy intakes: protein, carbohydrate (CHO) and fat, were reported in grams (g) and as a percentage of daily energy intake. Non-starch polysaccharide (NSP, dietary fibre), starch and total sugars were used to differentiate between CHO fragments [29] and total fat comprised of saturated fatty acids [35], mono-unsaturated fatty acids (MUFA), polyunsaturated fatty acids (PUFA) and trans fatty acids (TFA). Key micronutrients for pregnancy (dietary folate, iodine, vitamin D, calcium and Iron) were measured as milligrams (mg) or micrograms (µg). These specific nutrients were selected as being most associated with adverse pregnancy outcomes such as anaemia, pre-term delivery, low birth weight, pre-eclampsia and neurological defects [6,12,15,16,36,37,38,39,40], and therefore prioritised for analysis within this cohort. Mean values per participant for all nutrients recorded in the 3-day food diaries were determined, and a population average calculated. Data were compared to Dietary Reference Values (DRV) for pregnancy (where values have been determined) [41], specifically Estimated Average Requirements (EAR) for energy where half the population will require more than the EAR, and half less [41]. Specific micronutrients essential for optimum pregnancy outcomes were categorised into Reference Nutrient Intakes (RNI) and Lower Reference Nutrient Intakes (LRNI). The RNI is defined as two standard deviations above EAR and is deemed an adequate intake of a nutrient for 97.5% of a normally distributed population. Conversely, the LRNI is defined as two standard deviations below the EAR and represents the lowest intakes, which may be adequate for a very low number of individuals (2.5%), but inadequate for the vast majority [41].

Alcohol was not included in the analysis, as intakes were negligible. Further to this, participants were not asked to record consumption of dietary supplements, as NICE guidelines only routinely recommend folic acid (400 μg/day) and vitamin D (10 μg/day) supplements for pregnant women in the UK [42]. The study’s main aim was to evaluate the adequacy and quality of nutritional intakes and reflect dietary behaviour in line with dietary requirements/advice.

Nutrition data were analysed using a statistical package (SPSS, IBM SPSS Statistics Version 23 Armonk, NY, USA: IBM Corp) [43]. Data were cleaned and screened to check for outliers and explored to provide a numerical description of maternal characteristics and mean values for nutrient intakes. Inferential statistics were used to explore relationships between variables. One sample t-tests were used to compare intakes to DRV’s. Some data violated statistical assumptions, and therefore non-parametric tests Krushal–Wallis and the Friedman’s test were used to determine within-person variability between different time points and cohort differences in intakes.

Chi-square tests and independent t-tests were used to explore differences between groups according to categorical variables such as BMI and nutritional DRVs, [41]. A *p*-value < 0.05 was considered statistically significant. Dietary analysis considered the data primarily as one cohort considering nutritional status for the total sample of women living with obesity. Secondary analysis has investigated differences in dietary intakes relative to BMI (Section 3.4). However, to account for sample size, participants were categorised into 2 BMI subgroups groups, either women with a BMI 35–39.9 kg/m^2^ (*n* = 80) or women with a BMI ≥ 40 kg/m^2^ (*n* = 60).

## 3. Results

### 3.1. Participant Characteristics

Baseline data relating to age, BMI and maternal weight were collected for 140 women at approximately 11-weeks’ gestation (see Figure 1). Eligible women who declined to participate received routine antenatal care commensurate with their BMI.

Data were categorised according to BMI classification (Table 1). Over half of the women (57%) were categorised as obese class II (BMI = 35–39.9 kg/m^2^) whilst 6% had the highest levels of obesity (BMI ≥ 50 kg/m^2^), mean BMI 40 (SD 5.73) at booking-in. Three-quarters of the women in the study were aged 25–39, with 11% being over 40 years old; the mean age of the total sample was 30-years-old (SD 6.02). Nearly two-thirds of women were multiparous. Chi-square tests for independence revealed no statistical significance between BMI and parity or BMI and age. In total 13% of the sample classified themselves as current smokers. Chi-square tests for independence revealed no significant statistical associations between maternal characteristics (age, parity, BMI, smoking status), dietary intakes and pregnancy outcomes or birth weight (*p* > 0.05).

### 3.2. Clinical Outcomes

Of the 140 participants recruited into this study, 134 participants (4% attrition) remained in the study until completion with clinical outcome data collected from the women’s medical records (Table 2). In total, 8% had GDM, 9% hypertension and 12% had pre-eclampsia; 99% of women had live births. Over half (55%) had a vaginal delivery (48% spontaneous). A total of 19% of women had an elective caesarean, with 26% emergency caesarean. Low birth weight (defined as below 2.5 kg) was recorded for 7% of the babies born, whilst 5% were LGA (>4.5 kg). The majority of babies were born within the normal weight range (2.5–4.5 kg). The majority of babies did not require admission to Special Care Baby Unit, and APGAR scores at 1 (79%) and 5 min (96%) were considered normal (with scores between 7 and 10).

Weight change data at 36-weeks’ gestation was reported in clinical notes of only 39 women. GWG was less than 5 kg for more than (54%) of the women, with 18% of these women gaining less than 0 kg. Less than a quarter (23%) were recorded to gain weight within 5–9 kg, with 23% exceeded 9 kg GWG, who would be considered to have excessive GWG [8].

### 3.3. Dietary Analysis

Of the total sample recruited into this study, *n* = 140 women (100%) completed at least one 3-day food diary during the study period. In total, 66% and 71% of women completed the diary at time points 1 (T1) (16 weeks) and T2 (28 weeks), respectively. Although by 36-weeks’ gestation (T3), food diary submissions were provided by 52% of the total sample. A third of women repeated the food diaries at two-time points (T2 and T3), and 26% of women submitted all three food diaries (T1, T2 and T3).

#### 3.3.1. Macronutrients

The ratios of energy derived from the macronutrients, CHO and total fat were generally consistent with Estimated Average Requirements (EAR) [44], although intakes of total fat were significantly lower than EAR at time point 1 and CHO at time point 3 (see Table 3).

One-sample T-tests compared individual macronutrient intakes to dietary reference values (see Table 3). Results show that total sugars (T1, T2 and T3) significantly exceeded recommendations at all three time points and SFA at T2 and T3. Protein significantly exceeded recommendations at T2. By comparison, intakes of MUFA’s (at T1, T2 and T3) and PUFA’s (at T1, T2 and T3) significantly failed to achieve recommendations, and this was further highlighted by the ratio of PUFA to SFA (P:S ratio), which were significantly different to the recommended 0.8:1 at each visit (at T1, 2 and 3). Data also show that NSP (at T1, 2 and 3) and starch intakes (at T1, 2 and 3) failed to reach recommendations at a significant level. Non-parametric Friedman’s tests were conducted to determine within-person changes from the mean macronutrient intakes between time points 1, 2 and 3 (Table 3).

Dietary intakes remained relatively consistent throughout pregnancy, with intakes of total fat and carbohydrate being similar at each time point (*p* > 0.05), the exception being protein intake, which significantly increased between T1 and T2 and SFA, for which there was a highly significant increase in intakes during pregnancy.

#### 3.3.2. Micronutrients

Intakes relating to iron, folate, calcium, iodine and vitamin D intakes were analysed to determine the percentage of women achieving RNI or more and the percentage of women failing to achieve LRNI (see Table 4).

Table 4 shows a small proportion of women (14%, 10% and 19% at T1, T2 and T3, respectively), achieving the RNI of 14.8 mg/day for Iron, with most women not consuming the recommended intake throughout pregnancy. Overall, the pregnant women living with obesity recorded intakes of Iron which differed significantly throughout pregnancy compared to the RNI. A significant proportion (55–67% of women had iron levels throughout pregnancy only within the LRNI range (LRNI, specified as appropriate for only 2.5% of a population). A high proportion of women (31% at T1, 23% at T2 and 18% at T3) failed to achieve the LRNI of 8 mg/d for Iron, indicating inadequate intakes for the group.

Two-thirds to three-quarters achieved RNI for Calcium intake. Although noteworthy that 5.4% at T1 and 5.5% at T3 failed to achieve LRNI (above the accepted 2.5%). Whereas women achieving Calcium levels somewhere between LRNI and RNI ranged between 15 and 28% at T1, T2 and T3.

Intakes of iodine showed that approximately one-third of women achieved RNI at T1 and T2, and around 50% of women achieving RNI by T3. At T1 and T2, over half of the women, and at T3, 36% met the LRNI range. However, 18.3%, 13.1% and 8.2% were below the LRNI of 69.9 µg/day at T1, T2, and T3 (respectively).

Intakes of dietary folate presented a third to a quarter of women who achieved the RNI of 300 µg/day., although 66–74% achieved levels above the LRNI but below RNI.

Consumption of vitamin D (10 µg/day), as expected, was not achieved via diet intake for the majority of women (96.8%+).

### 3.4. BMI and Dietary Composition

Chi-square tests revealed no significant statistical associations between micronutrient intakes and BMI subgroups. Kruskal–Wallis tests were used to explore whether differences in BMI impacted intakes of micronutrients. There were no significant differences in intakes for vitamin D, Calcium, Folate, Iodine or Iron at T2 and T3. However, there was a statistically significant difference in iron intakes at T1 across BMI groups, with participants with a BMI ≥ 50 kg/m^2^ having lower intakes than participants with a BMI 40–44.9 kg/m^2^. Further to this, Friedman tests were conducted to compare intakes of micronutrients between T1, T2 and T3 within the cohort; however, there were no significant differences, possibly due to the limited sample size.

## 4. Discussion

This study recruited a sample of pregnant women living with obesity and followed them through pregnancy to consider their obesity status, GWG, nutritional intake and associated clinical outcomes.

Our findings report adverse clinical outcomes, consistent with previous literature [6,28,39,45,46,47] and compared to the general pregnancy (all data) population, for which the status of obesity itself might explain such outcomes. Although not specifically GWG, contrary to previous research [45,48] in this study, the majority of women (77%) (whose weight change was calculated) did not gain more than the IOM (2009) GWG recommendations [9], despite living with obesity. In line with previous evidence [45], increased BMI was not an indication of (excessive) GWG. The sample size of GWG findings was underpowered, however the results would support the UK’s approach [2] in promoting a healthy diet for pregnant women. However, the findings indicate that many pregnant women did not meet the recommendations for a range of nutrients essential for maternal health and foetal development, suggesting that pregnant women living with obesity have variable quality diets and may require specific guidance and tailored nutritional support. The following discussion provides consideration of the findings in this study relevant to the pregnant women’s dietary intake; of macro and micro-nutrients and the recorded clinical outcomes for this group of women living with obesity.

### 4.1. Diet Quality

At first glance, the overall diet quality of the pregnant women living with obesity looks satisfactory. However, when evaluating dietary composition, the women showed varying levels of diet quality.

#### 4.1.1. Macronutrients

Regarding macronutrient composition, total energy intake did not differ significantly from the EAR. Similar to NDNS findings [10], protein intake was significantly higher than the EARs at time point 3. Further to this, European guidelines suggest that a protein intake of 10–25% of energy intake appears to be safe, and so the intakes recorded can be considered moderate and comparable to protein intakes recorded in the WATCH study [17]. Total carbohydrate intake did not change significantly throughout pregnancy and achieved the EAR, although, in the third trimester, there was a significant reduction from EAR, though again here would be considered within an acceptable range. In line with the healthy diet approach of the eat well guide [29], carbohydrate intake should be consumed through starchy foods, with low glycaemic index and high in dietary fibre, such as vegetables, legumes, fruits and whole grains. Further breakdown of carbohydrates composition, however, suggests the carbohydrate intake in the women’s diets was poor quality, as sugar intake was significantly above EAR (>5% of energy) throughout pregnancy; consistently recorded at more than double the recommended intake, and in contrast, significantly lower levels than recommended levels (<30 g/day) were recorded for dietary fibre. High added sugar and low dietary fibre intakes are associated with several non-communicable diseases, such as dental caries, obesity and bowel disorders such as constipation, diverticular disease and increased risk of colorectal cancer [49].

Total fat intake was within the 35% energy EAR range and did not differ significantly throughout pregnancy. However, it is noteworthy that in Europe, total fat intake recommendation has a range from 20 to 35%, which would suggest in comparison that the total fat intake of women in this study consistently measured towards the higher acceptable range. However, further breakdown of fat intake shows SFA as significantly above EAR, and SFA intake also increased significantly throughout pregnancy. High intakes for SFA are associated with poor cardiometabolic health [50]. In contrast, significantly lower EAR levels were recorded for MUFA and PUFA. PUFAs are considered the essential fatty acids for foetal growth and development and health throughout the life course, whereas SFAs are synthesised by the body and are not required in the diet, though dietary recommendations acknowledge up to 10% of energy intake as acceptable [44].

#### 4.1.2. Micronutrients

The findings suggest that many pregnant women are not meeting the UK recommendations for a range of micronutrients essential for positive clinical outcomes for both the mother and her offspring. The findings suggest dietary intakes of very low levels of Iron, with a significant deviation from the RNI, throughout pregnancy. Only 10–19% of women achieved the RNI of 14.8mg daily at any one time point. A significant majority of women (55–67%) had iron intakes throughout, at the LRNI, although up to 31% of women did not achieve the LRNI of 8mg per day. For this population to be considered as having adequate iron intakes, only 2.5% of women would be at LRNI [41], so 31% below LRNI indicates a very high level of inadequacy. Iron deficiency is extremely prevalent globally, particularly among women of childbearing age; hence the use of routine iron supplementation during pregnancy in several countries [6]. Dosage of supplementation varies widely in different countries (ranging from 9 to 50 mg/day), with WHO (2016) recommending 27 mg/day. However, iron supplementation is not routinely offered to pregnant women in the UK [51].

In the UK, the RNI for Iodine during pregnancy is 140 μg/day [15,41]. In this study, one-third to half of the women achieved this intake, with a significant proportion of women at the LRNI, and of concern 8–18% not meeting the LRNI of 70 mg/day, again showing high levels of inadequate intakes in this population. Only small amounts of iodine (150 µg/day) are required to prevent deficiency. It is noteworthy that in Europe, the RNI for iodine is higher than in the UK at 250 µg/day [15].

Vitamin D is only found in a few foods, including oily fish or fortified dairy products. Dietary intake did not meet daily intake recommendations, and this is similar to the general population of women (childbearing age) [52]. The primary source of Vitamin D is the action of UV light on the skin, hence the UK recommendation for vitamin D supplementation of 10µg per day for all adults in the population and throughout pregnancy and lactation [2,13]. However, it highlights the substantial risk of deficiency in this group, should supplements be absent or not taken regularly.

In the UK, the RNI for folate intake during pregnancy is 300 µg [41], in this study at any one time point, only a quarter to one-third of the women achieved the recommended intake, with over two-thirds of women at the LRNI. A 400 ug folic acid supplement is recommended for all pregnant women in the first trimester and preferably 3 months before conception to reduce the risk of deficiency [42].

The micronutrient evaluation of dietary composition in women living with obesity in this study is concerning. The LRNI values indicate an acceptable nutritional intake for only 2.5% of a population with low requirements. In this sample, a high percentage of the women living with obesity were consistently at the LRNI, and a further percentage were not meeting these lower recommended intakes, indicating marked sub-optimal nutrition. Insufficient intake of these nutrients is associated with a range of severe adverse pregnancy outcomes, including preeclampsia, low birth weight and poor cognitive development [39,46,47]. Folate deficiency can increase the risk of adverse outcomes, including neural tube defects (in early pregnancy), preeclampsia and low birth weight, and Vitamin D deficiency may also contribute to these, as well as GDM. Previous research has highlighted that nutritional deficiencies may contribute to low Apgar scores, particularly Iron. Even a mild deficiency of iodine during pregnancy is associated with impaired IQ in children, and optimal iodine intakes do not improve this adverse effect during childhood [46]. Suboptimal maternal nutrition is also associated with poor foetal growth and development, plus increased risk of non-communicable disease later in life [53,54]. For both preeclampsia and GDM, prevention and treatment are promoted by healthy diets, and prevalence has been linked to various macro-micro nutrient deficiencies.

This study concurs with WHO guidelines [6], highlighting that the higher the BMI in pregnancy, the greater the risk of micronutrient deficiencies. Overall, the findings suggest sub-optimal dietary macro and micro-nutrient intakes, which may have contributed to adverse clinical outcomes in this sample.

### 4.2. Clinical Outcomes

Considering the obstetric outcomes of the women in this study, over half of the women had a vaginal delivery (49% spontaneous; 8% instrumental), similar to that of the general population pregnancy delivery statistics (52% spontaneous, 12% instrumental) [55]. In line with previous research, the percentage of women who had (elective 19%/emergency 26%) caesarean was higher than the general population statistics (elective 12.6%; emergency 16.4%, [56]. The recorded Apgar score 5 min after birth was below normal for 4% of babies, compared to 1.4% of the general population.

Preeclampsia affects 3–5% of pregnancies in the general population, though in this study, preeclampsia occurred in 12% of the sample. Preeclampsia is of interest to foetal outcomes of birth weight. In this sample, birth weight (low birth weight at 7%) and admissions to special care baby units were in line with all pregnancy population data [55].

Prevalence of GDM was recorded at 8% in this study, in comparison the UK, the rate is around 4% [57], although individual characteristics, such as ethnicity, increase the incidence; prevalence ranging between 1·2 and 8·7% in White British women, compared to 4–24% in South Asian women in the UK). This study was conducted in part of the UK, with low ethnic diversity, with 88.8% of the general population being white. Therefore, a GDM of 8% may be considered at the upper levels of GDM prevalence within the sample.

### 4.3. Implications for Practice

Pregnant women are advised to consume a healthy diet, and this study has evaluated actual diet composition in pregnant women living with obesity. The WHO (2020) state that “malnutrition refers to deficiencies, excesses, or imbalances in a person’s intake of energy and/or nutrients” [58]. This study reports sub-optimal diet consumption for a range of macro-and micro-nutrients in the study sample. It has been previously recognised that many healthcare professionals would fail to recognise malnutrition in individuals with obesity [59] though limited research has specifically explored diet quality in pregnant women living with obesity.

Midwives are the interface between clinical guidelines and pregnant women, but there is a lack of confidence and expertise in delivering nutritional advice to pregnant women, particularly women living with obesity [7]. Nutrition education needs to be more explicitly embedded into midwifery training [60]. The RCOG (2018) suggests pregnant women living with obesity to be referred and supported by a dietitian [2], though women with obesity report rarely receiving support [7]. Midwives should refer pregnant women to a dietitian as required. Care pathways need revising so that women living with obesity are supported throughout pregnancy to optimise their nutritional composition, to improve pregnancy and clinical outcomes.

A systematic review focused on dietary interventions [21] has highlighted the predominance of holistic lifestyle interventions during the antenatal period that exists to improve pregnancy outcomes, manage GWG, reduce incidences of GDM and large for gestational age babies. The review noted the wide range of dietary advice, which was generally based on healthy eating recommendations, with only two studies offering pregnancy-specific advice. Studies have shown that improving maternal eating behaviours can reduce GWG, improving maternal inflammatory markers [61], reduce adipose tissue inflammation in the offspring [62] and influence positive weaning and early child feeding practices in their offspring [61,63]. However, there is limited research targeting the improvement of diet quality in pregnant women living with obesity. However, previous research on diet interventions in pregnancy, though not explicitly targeting women living with obesity, has shown some improvements in diet quality and pregnancy outcomes. Another small-scale study that focused on improving the dietary quality of pregnant women living with obesity showed promising outcomes.

Providing evidence-based and effective nutritional interventions to pregnant women living with obesity may not only demonstrate better clinical outcomes, but it is also likely to be cost-effective, reducing healthcare costs of treating associated conditions such as preeclampsia, GDM or ongoing medical and developmental needs of the new-born. Although further research is needed. All pregnant women living with obesity should receive personalised and well-designed dietary interventions. These findings suggest the need for interventions to reduce SFA and sugar intake and promote diets rich in food sources of key micronutrients (particularly Iron and Iodine).

### 4.4. Future Research

Individual requirements for micronutrients have previously been defined as ‘the amount required to prevent clinical signs of deficiency [41]. However, it is unclear whether obesity increases requirements for some nutrients [18,64], requiring further investigation.

As previous research [45] suggests women, living with healthy, overweight or class I obesity may also be at risk of excessive GWG or dietary composition limitations. Therefore, future research is warranted to evaluate the dietary make-up in these groups. It would also be helpful to explore other demographics such as the role of socioeconomic status, race, the prevalence of smoking, age, parity and supplement use regarding dietary intake and pregnancy outcomes in this population. Furthermore, specific evaluation of women living with (or lived with) obesity who have previously undergone bariatric surgery are a particularly increased risk for nutritional deficiencies during pregnancy (due to insufficient intakes and/or poor absorption). Although limited, some research suggests women are at increased risk of low Iron, vitamin B12, folate and fat-soluble vitamins, and deficiencies in macros-nutrients (fat and protein) [65] and further evidence on this target sample is warranted.

### 4.5. Strengths and Limitations

This study has assessed direct dietary intake given the assumption that women in pregnancy should be encouraged to consume a healthy diet. Understanding consumption through real-world application and not interfering with diet or healthcare practices can be considered a strength of this observational study. The study targeted and recruited a specific cohort of women living with obesity, considered a high-risk pregnancy, for which there is limited previous research evidence on this population group. However, the analysis has not considered the biological mechanisms of dietary requirements, such as factors affecting the absorption of nutrients (e.g., high calcium intake may reduce iron absorption), moreover, the analysis has not accounted for dietary supplements, for which there are two recommended in the UK, Vitamin D and Folic acid. However, women in the general population and those who are pregnant and living with obesity are less likely to take nutritional supplements [66], highlighting the need to consider diet consumption as a reflection on the lives of pregnant women living with obesity.

This study was an extension of an established FFB trial, and therefore the recruitment protocol was restricted to this opportunistic sample, recruiting from a set pool of women living with obesity class II and above (BMI ≥ 35 kg/m^2^). Whilst there was a range of BMI (35 to 50+ kg/m^2^), the participant characteristics were not significantly different across the cohort. Analysis was conducted to assess obesity, nutritional and clinical outcomes. The further breakdown into sub-obesity classifications was not applied due to variable sample sizes. It is noteworthy that of the 140 women recruited, clinical outcome data were recorded for 134 women, representing 96% of the original sample. However, the study power is limited by participant attrition for GWG and dietary intake data. We acknowledge that the measurement of weight was only recorded in 39 women’s medical records, and therefore the calculation of GWG change and subsequent analysis of dietary composition in response to excessive GWG was not feasible. Whilst the study could have sought self-reported GWG, or the mother’s weight at delivery, this was not implemented, given women have been found to under-report weight at delivery and over-report GWG, which would have reduced the reliability of data. However, the lack of recorded weight in health records highlights an additional issue in maternity care, relating to the recommendation to weigh women during later pregnancy weeks (for birth assessment), and this may be highlighting an issue with clinical practice that requires further investigation.

Whilst all women submitted a 3-day food diary at some point of the study, the number of women who submitted across specific time points varied from 71 to 52%. Furthermore, the number of women who submitted two (at Time points 1 and 3) or three diary entries (at Time points 1, 2 and 3) was 33 or 26 percent, respectively. The 3-day food diaries were chosen as the best method for dietary collection for this study in recognition that other methods (such as food frequency questions are retrospective and may be less reliable). A strength of the method employed included the appointment to clarify and validate the quality of their diet records using various tools, which we suggest increased the accuracy and reliability of reported dietary intake and enhanced the subsequent nutritional analysis. However, it is noteworthy that for future research, online food diaries and research appointments may increase submission.

## 5. Conclusions

Women living with obesity are often considered to be adequately or even over nourished. However, data analysed from the women in this study suggest variable diet quality. Women with obesity may present with a sub-optimal diet, with a high intake of energy from saturated fatty acid and sugar and consistently lower intakes of ‘desirable’ macronutrients (fibre/starch/PUFA). Moreover, pregnant women with obesity may be more likely to consume sub-optimal intakes of essential micronutrients, and together nutritional status may contribute to increased risk for maternal and foetal outcomes. This study may help understand the higher rates of preeclampsia, GDM, low birth weight and high Apgar scores in this sample.

The UK’s current approach, via national guidelines [57,67], targets obesity as the origins of adult-related non-communicable diseases, failing to address the fundamental influence of nutritional quality on disease risk. Obesity is a consequence of suboptimal nutrition and not just the intake and storage of excess energy. To address the intergenerational transmission of poor health via poor quality nutrition warrants a multi-disciplinary and multi-modal approach to reverse the trend. A focus away from ‘dieting’ and weight onto positive, healthy eating messages that emphasise the critical nutrients required for good maternal outcomes and healthy development of unborn children needs urgent prioritisation for this group of women in the UK.

## Figures and Tables

**Figure 1 nutrients-13-01652-f001:**
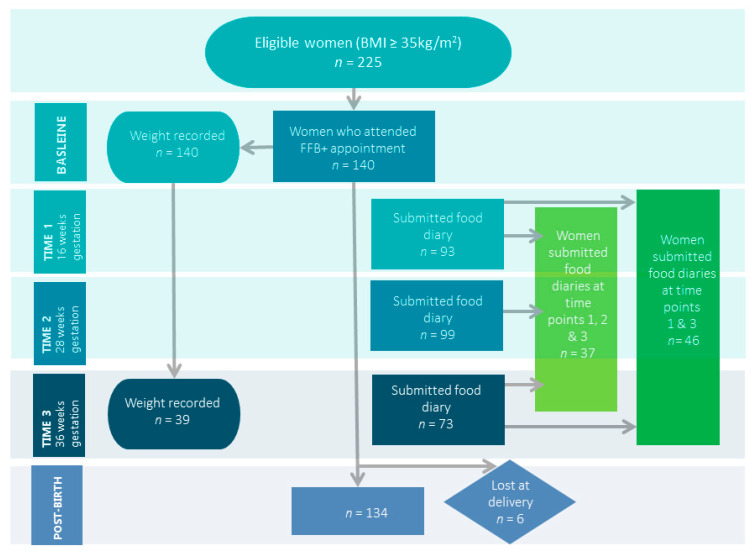
Participant recruitment and data collection.

**Table 1 nutrients-13-01652-t001:** Participant characteristics, according to BMI classification.

	Body Mass Index * *n* = 140				
	35–39.9 *n* = 80 (57%)	40–44.9 *n* = 37 (26%)	45–49.9 *n* = 15 (11%)	50+ *n* = 8 (6%)	All BMI (100%)
Age (*n* = 140)					
18–24 25–29 30–39 40+	15 (11) 32 (23) 27 (19) 6 (4)	4 (3) 13 (9) 16 (11) 4 (3)	4 (3) 6 (4) 4 (3) 1 (1)	1 (1) 4 (3) 3 (2) 0	24 (17) 55 (39) 50 (36) 11 (8)
Parity (*n* = 140)					
Primip (1st pregnancy) Multip (2nd + pregnancy) ^§^ Missing	32 (23) 47 (34) 1 (1)	10 (7) 27 (19) 0	6 (4) 8 (6) 1 (1)	5 (4) 3 (2) 0	53 (38) 85 (61) 2 (2)
Smoking status (*n* = 140)					
Given up Non-smoker Smoker Missing	9 (7) 62 (44) 8 (6) 1 (1)	7 (5) 24 (17) 6 (4) 0	0 10 (7) 4 (3) 1 (1)	1 (1) 7 (5) 0 0	17 (12) 103 (74) 18 (13) 2 (1)

Key: number of participants within the range (% of participants within the range). * Body mass index = kg/m^2^; ^§^ Multip 2nd + (subsequent pregnancies).

**Table 2 nutrients-13-01652-t002:** Clinical Outcomes.

	Body Mass Index				
	35–39.9	40–44.9	45–49.9	50+	All BMI
	*n* = 80 (57%)	*n* = 37 (26%)	*n* = 15 (11%)	*n* = 8 (6%)	−100%
Weight gain at 36 weeks (*n* = 39)					
Less than 0	2 (5)	4 (10)	1 (3)	0	7 (18)
0–4.9 kg	9 (23)	4 (10)	1 (3)	0	14 (36)
5–9 kg	7 (18)	2 (5)	0	0	9 (23)
9 kg+	7 (18)	1 (3)	1 (3)	0	9 (23)
Birth weight kg (*n* = 134)					
0.1–2.49	6 (3)	1 (1)	1 (1)	1 (1)	9 (7)
2.5–4.49	67 (50)	33 (25)	12 (9)	6 (4)	118 (88)
4.5–6.00	3 (2)	2 (2)	1 (1)	1 (1)	7 (5)
Hypertension (*n* = 134)					
Yes	7 (5)	1 (1)	2 (2)	2 (2)	12 (9)
No	69 (52)	35 (26)	12 (9)	6 (5)	122 (91)
Pre-eclampsia (*n* = 134)					
Yes	10 (8)	3 (2)	3 (2)	0	16 (12)
No	66 (49)	33 (25)	11 (8)	8 (6)	118 (88)
Gestational Diabetes 28 weeks (*n* = 134)					
Yes	6 (5)	4 (3)	1 (1)	0	11 (8)
No	70 (52)	32 (24)	13 (10)	8 (6)	123 (92)
Birth outcome (*n* = 134)					
Live birth	75 (56)	36 (27)	14 (11)	8 (6)	133 (99)
Stillborn	1 (0.8)	0	0	0	1 (1)
Induction of labour (*n* = 134)					
No	52 (39)	24 (18)	8 (6)	5 (4)	89 (66)
Yes	24 (18)	12 (9)	6 (5)	3 (2)	45 (34)
Mode of delivery (*n* = 134)					
Spontaneous	39 (30)	16 (12)	7 (5)	3 (2)	65 (48)
Instrumental	6 (5)	2 (2)	2 (2)	0	10 (7)
Elective caesarean	13 (10)	8 (6)	2 (2)	2 (2)	25 (19)
Emergency caesarean (EmC)	14 (11)	8 (6)	3 (2)	2 (2)	27 (20)
EmC with failure to progress	4 (3)	2 (2)	0	2 (2)	8 (6)
Admitted to Special Care Baby Unit (*n* = 133)					
No	67 (50)	34 (26)	13 (10)	7 (5)	121 (91)
Yes	8 (6)	2 (2)	1 (1)	1 (1)	12 (9)
APGAR score 1 min (*n* = 133)					
Critically low 0–3	3 (2)	3 (2)	1 (1)	0	7 (5)
Low 4–6	11 (8)	4 (3)	4 (3)	3 (2)	22 (17)
Normal 7–10	61 (46)	29 (22)	9 (7)	5 (4)	104 (78)
APGAR score 5 min (*n* = 133)					
Low 4–6	2 (2)	3 (2)	0	0	5 (4)
Normal 7–10	73 (55)	33 (25)	14 (10)	8 (6)	128 (96)

Key: number of participants within the range (% of participants within the range).

**Table 3 nutrients-13-01652-t003:** Macronutrient intakes as a percentage of estimated average requirements for energy.

	Dietary Reference Values (DRV)	Time 1 (16–20 Weeks) (*n* = 93)	Time 2 (28 Weeks) (*n* = 99)	Time 3 (36 Weeks) (n = 73)	Changed over Time (*n* = 37)
Macronutrient	EAR for energy	Mean ± SD % (EI)	Mean ± SD % (EI)	Mean ± SD % (EI)	
Total Energy (kcals)	1945 kcal T1 and T2 2145 kcals T3	1849 ± 591 95%	1984 ± 526 102%	2066 ± 587 93%	*p* > 0.05
Protein %E	15% of EI	15.8 ± 3.0 106%	16.2 ± 4.4 108% ^1^	14.8 ± 4.4 98.4%	*p* = 0.031
Total Fat %E	35% of EI	33.4 ± 6.8 95% ^2^	34.8 ± 6.2 99%	35.7 ± 6.2 102%	*p* > 0.05
SFA %E	11% of EI	12.0 ± 3.3 109%	12.9 ± 3.2 117% ^1^	13.3 ± 3.2 121% ^1^	*p* = 0.0015
MUFA %E	13% of EI	10.7 ± 3.1 82% ^2^	11.3 ± 2.9 87% ^2^	11.5 ± 2.6 88% ^2^	*p* > 0.05
PUFA %E	6.5% of EI	5.6 ± 2.4 86% ^2^	5.65 ± 2.1 87% ^2^	5.8 ± 2.1 89% ^2^	*p* > 0.05
P:S Ratio	0.8:1	0.51 ± 0.26 64% ^2^	0.47 ± 0.23 59% EAR ^2^	0.46 ± 0.21 58% ^2^	*p* > 0.05
CHO %E	50% of EI	50.7 ± 7.5 101%	48.9 ± 6.6 98% EAR	47.3 ± 6.8 95% ^2^	*p* > 0.05
NSP * g/day	18g/day	12.8 ± 5.2 71% ^2^	12.8 ± 4.1 71% EAR ^2^	12.8 ± 4.1 71% ^2^	*p* > 0.05
Sugars %E	5% of EAR	23.5 214% ^1^	24.9 226% EAR^1^	22.5 205% ^1^	*p* > 0.05
Starch %E	39% of EAR	26 67% ^2^	27 69% ^2^	28 72% ^2^	*p* > 0.05

Key DRV–Dietary Reference Values; EAR–Estimated Average Requirements; MUFA–Mono-unsaturated Fatty Acids; PUFA–Polyunsaturated Fatty Acids; P:S ratio–polyunsaturated fatty acid: saturated fatty acid; CHO–carbohydrate; NSP–Non-starch Polysaccharide; RNI–Reference Nutrient Intake; EI–Energy intake;. ^1^ Significant difference where nutrient intakes exceed DRV *p* < 0.05. ^2^ Significant difference where nutrient intakes fail to meet DRV *p* < 0.05. * NSP major component of dietary fibre and used as the analysis was before SACN update in 2015.

**Table 4 nutrients-13-01652-t004:** Percentage of women achieving RNI and LRNI for pregnancy-related micronutrients.

Micronutrient and (UK Recommended Intake)	DRV	Ranges	Time 1 % Achieved	Time 2 % Achieved	Time 3 % Achieved
Iron (14.8 mg)	<LRNI LRNI ≥ RNI	<7.99 8.0–14.7 ≥14.8	31.2 54.8 ^2^ 14.0	23.2 66.7 ^2^ 10.1	17.8 63.0 ^2^ 19.2
Calcium (700 mg)	<LRNI LRNI RNI	<399.9 400–699.9 ≥700	5.4 28.0 ^1^ 66.7	2.0 23.2 ^1^ 74.7	5.5 15.1 ^1^ 79.5
Iodine (140 μg)	<LRNI LRNI RNI	<69.9 70–139.9 ≥140	18.3 50.5 ^2^ 31.2	13.1 51.5 ^1^ 35.4	8.2 35.6 56.2
Vitamin D (10 μg) *	<RNI RNI	<9.99 ≥10	96.8 ^2^ 3.2	98.0 ^2^ 2.0	98.6 ^2^ 1.4
Folate (300 μg)	<LRNI LRNI RNI	<99.9 100–299.9 ≥300	1.1 66.7 ^2^ 32.3	0 73.7 ^2^ 26.3	0 65.8 34.2

Key DRV—Dietary Reference Values; RNI—Reference Nutrient Intake, LRNI—Lower Reference Nutrient Intake. * LRNI for vitamin D not available. ^1^ Significant difference where nutrient intakes exceed DRV *p* < 0.05. ^2^ Significant difference where nutrient intakes fail to meet DRV *p* < 0.05.

## Data Availability

The data are not publicly available due to ethical limitations publishing medical record data, which may expose individuals as identifiable.

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
