# Peer review of "Pregnant Women Living with Obesity: A Cross-Sectional Observational Study of Dietary Quality and Pregnancy Outcomes"

_nutrients, 2021, doi:10.3390/nu13051652_

Round 1

Reviewer 1 Report

Introduction:

Overall comment:  Well written and interesting paper.  the introduction is too long!  Excessive information on gestational weight gain, especially since your study has this information on only 39 women.  Seems to me that paragraph lines 72-85 are unnecessary.  It should, however, include global and UK recommendations for dietary supplements since for some of the nutrients you target, supplementation is recommended during pregnancy. 

Line 42: please add acronym RCOG since that is used subsequently but has not been previously defined.

Line 63: reference 8 has been updated  in 2013, please update reference and related content

Line 101: you include food sources of folate.  Folate fortification of foods is common in many countries which deserves a mention because women or reproductive age are known to have low folate intake.  If not included here, it should be included in the discussion.

Line 124: please specify the “key nutrients” included in your hypothesis. And why you limited it to those 5.

Materials and Methods

Line 134:  Please provide the reader with more information on the population and how they were recruited – when?, approached by how many midwives?, where? No age cut-off for “older mothers”? No demographic info?  How was the sample size determined? What is the power?

Line 137-138 : unclear relevance or meaning and is an incomplete sentence.

Line 148: methods indicate weight was recorded during midwife care appointments, but was not…was this a prospective study and midwives aware of study protocol?  Why was it not recorded?

Line 157: did women measure or weigh their foods? Why not?  This is a limitation so please include in the limitations that portion sizes were estimated and the potential for error

Line 176: give information on how you validated to determine reliability of records

Line 173:how did the approximation of wastage, peeling, seeds occur.  You note it was a “researcher”.  Was this person trained to do this or what was their culinary expertise to accurately do this adjustment of the weight/size of the serving?

Line 176: who makes Microdiet, version and is it validated?

Line 183: were mean diet values calculated per women? and then averaged?  Please clarify.  Did you assess the within person variability in intake – that should be included.  Were there any outliers? For any nutrients?

Line 188: are EARs or RNIs established for pregnancy?  Please clarify if that is what was used

Line 195: did you ask about Micronutrient supplement intake? If not say, so and that is a limitation as well to discuss.  You suggest that these low intakes are associated with poor birth outcomes, but maybe it is just obesity and inflammation or older age (8%) or smoking (13%) since you do not know if they in fact had reduced intake of the nutrient if supplementation information is not included. What were birth outcomes in the smokers and >40 yr women? In many parts of the world folate and iron are the minimum recommended supplements during pregnancy; in other placed multiple MN supplementation during 2nnd and 3rd trimester are the standard of care.

Line 210: with 2 BMI categorizations, what was your ability to detect a difference between 2 groups with this sample size (and since you found no difference in 3.4 section overall, what would your sample size needed to be to detect a difference?)

Any reason you excluded so many other micronutrients since most analysis software includes analyses for most micronutrients.  You could at least comment on whether other micronutrients were of concern.

Results

Figure 1 – what happened to the 85 women who did not enroll, please include this info; suggest you also include birth outcomes in the post birth section

Line 237 and throughout: p-values, I prefer an actual value rather than <0.05, but I will leave that the editor.

Line 245:  I disagree that there were relatively few pregnancy complications – just report ; also what about rate of macrosomia?

Line 263: any association between GWG and age or parity?

Table 3 seems to be missing parts of the headers; it was difficult to read; why not include % not meeting LRNI?; %exceeding RNI)?; protein – why no mean pro g at time points but given the RNI.  Also indicate which are EAR and which are RNI; for sugar and FSA, include how many gm/day footnote should include all abbreviations

What happened to Table 4? Check table numbering

Table 5:Vit D no LRNI?

Discussion

Lines 340-342: please cite “previous literature” and be more specific to outcomes you are talking about

Line 346-348: based in your study you cannot make this conclusion.  You had only 39 people with GWG so cannot extrapolate to the population.  Dismissing specific wt guidance is also not the aim of the study since specific guidance was not assessed.  Concur with promoting healthy diet cannot be argued against, but is not defined, so that is not helpful.  Suggest you stick to what you measured with an adequate sample size, since there is info on diet guidance that would be helpful given what you found about macronutrients.

Line 364: Unclear why you could not look at the food records and determine proportion of servings coming from high biological value (you could report pro servings in terms of animal sources if you wanted); regardless since so many met the pro goal, uncertain how much this is relevant at all, so either add info or delete.

For the whole micronutrient section, need to compare your findings against other studies of obese pregnant women; or compare to obese non-pregnant women if not available in pregnancy.

Line337: include consequences of high SFA and added sugars, inadequate fibre in pregnancy

Line 391: compare your findings to other studies in preg women

Line 394: low fe intake is common during pregnancy, hence need for supplementation – need to include information on this guidance; same for Line 421 on folate – include rationale for supplementation worldwide

Line 432: look for a relationship between preeclampsia (12%) and LBW (7%) and your variables.  What were the characteristics of those women?  In terms of obesity class, age, dietary intake, GWG?

Line 462: characteristics Typo?

Line 483: since so few women were referred to a dietitian, add this is an opportunity for midwives to make referrals

Lines 512-513 – excellent point!

Future research should also include role of socio-economic status, ,race,  smoking, education, parity, inflammation, dietary advice/counselling, referral to dietitian, age on dietary intake and GWG and birth outcomes; does micronutrient supplementation mitigate worse birth outcomes.

Author Response

Response to reviewer 1:

Reviewer 1 comments

Author Response

Overall comment:  Well written and interesting paper. 

Thank you

The introduction is too long!  Excessive information on gestational weight gain, especially since your study has this information on only 39 women. 

Seems to me that paragraph lines 72-85 are unnecessary. 

It should, however, include global and UK recommendations for dietary supplements since for some of the nutrients you target, supplementation is recommended during pregnancy. 

We have shortened the introduction and removed some of the information about GWG.

We have removed lines 72-85.

We have added information about WHO (2016) and UK recommendations for dietary supplements during pregnancy. See line numbers 84 - 87

Line 42: please add acronym RCOG since that is used subsequently but has not been previously defined.

amended

Line 63: reference 8 has been updated  in 2013, please update reference and related content

Amended now line 64

Line 101: you include food sources of folate.  Folate fortification of foods is common in many countries which deserves a mention because women or reproductive age are known to have low folate intake.  If not included here, it should be included in the discussion.

We have included information about mandatory folic acid fortification of flour. See line numbers 87 - 89

Line 124: please specify the “key nutrients” included in your hypothesis. And why you limited it to those 5.

Reference to key nutrients and reason for selecting have been made clearer in the methods section. See lines197 - 202

Materials and Methods

Line 134:  Please provide the reader with more information on the population and how they were recruited – when?, approached by how many midwives?, where? No age cut-off for “older mothers”? No demographic info?  How was the sample size determined? What is the power?

This has been rephrased to reflect reviewer’s comments; however, the exclusion criteria did not include a maximum age for participation. The exclusion criteria has been corrected see lines 141 - 146

Acknowledgement of the sample size limitations has been added into the methodology see lines 128 – 133

& in results: “however, there were no significant differences, possibly due to the limited sample size” (see lines 356 - 357.)

As this was a subsample of a larger study Narayanan et al.2016 [27]. We did not conduct a power calculation for the anticipated sample size. As a result  the analysis was conducted using non-parametric tests to mitigate this limitation somewhat, and we acknowledge sample size of similar studies e.g. Jones et al.2010 [36].

A comment in the discussion recognising the sample size required for dietary analysis and GWG has been added. See lines 369 – 371

We have added further commentary into the discussion study limitations to reflect the sample size and lack of power. Lines 568-579

Line 137-138 : unclear relevance or meaning and is an incomplete sentence.

Rephrased and is clearer. See lines 123 - 125

Line 148: methods indicate weight was recorded during midwife care appointments, but was not…was this a prospective study and midwives aware of study protocol?  Why was it not recorded?

More detail added with regards to recruitment process and sampling strategy. This study opportunistically sampled participants through the FFB cohort, specifically recruiting those women referred into obstetric services for additional antenatal support as deemed a high-risk pregnancy see lines 128 - 133

Line 157: did women measure or weigh their foods? Why not?  This is a limitation so please include in the limitations that portion sizes were estimated and the potential for error

This has been rephrased to be more explicit. See lines 156 -158

Line 176: give information on how you validated to determine reliability of records

Clarification on the methodology and validation of data coding has been added, specifically data was checked by the supervising and senior dietitian (last author)

See lines 190 - 191

Line 173:how did the approximation of wastage, peeling, seeds occur.  You note it was a “researcher”.  Was this person trained to do this or what was their culinary expertise to accurately do this adjustment of the weight/size of the serving?

We have clarified the process and updated the text to acknowledge the researcher was trained in this method. See lines 172 & 185 - 187

Line 176: who makes Microdiet, version and is it validated?

An appropriate reference has been added. See line 188

Line 183: were mean diet values calculated per women? and then averaged?  Please clarify.  Did you assess the within person variability in intake – that should be included.  Were there any outliers? For any nutrients?

Rephrased see line 202 - 204 and a sentence added in methods re cleaning and screening data for outliers. See line 221

Line 188: are EARs or RNIs established for pregnancy?  Please clarify if that is what was used

This has been clarified in text  see lines 204 - 205

Line 195: did you ask about Micronutrient supplement intake? If not say, so and that is a limitation as well to discuss. 

You suggest that these low intakes are associated with poor birth outcomes, but maybe it is just obesity and inflammation or older age (8%) or smoking (13%) since you do not know if they in fact had reduced intake of the nutrient if supplementation information is not included. What were birth outcomes in the smokers and >40 yr women?

In many parts of the world folate and iron are the minimum recommended supplements during pregnancy; in other placed multiple MN supplementation during 2nnd and 3rd trimester are the standard of care.

Rephrased to make more explicit. We did not record supplement intake as the aim of the study was to assess actual diet intake according to dietary recommendations specifically. See lines 215 - 219

Thank you for this suggestion we have conducted additional analysis here.  Statistical testing revealed no statistically significant associations between smoking and age with pregnancy outcomes and birthweight. A sentence has been added to reflect this. See line 250 - 252

We have added a comment re NICE guidelines and recommendations for folic acid and vitamin D. see lines 215 -217.  Further information about this is also in the discussion. See lines 442 & 447

Line 210: with 2 BMI categorizations, what was your ability to detect a difference between 2 groups with this sample size (and since you found no difference in 3.4 section overall, what would your sample size needed to be to detect a difference?)

Non-parametric analysis was conducted to account for the sample size limitation. A comment on acknowledgment of sample size has been added to the discussion. See lines 586 - 588

Any reason you excluded so many other micronutrients since most analysis software includes analyses for most micronutrients.  You could at least comment on whether other micronutrients were of concern.

The selection of micronutrients related to their association with adverse pregnancy outcomes has been emphasised and most notable adverse outcomes have been referenced see lines 199 - 202

Results

Figure 1 – what happened to the 85 women who did not enroll, please include this info; suggest you also include birth outcomes in the post birth section

Have made a comment that women who declined to participate received antenatal care appropriate to their BMI. See line 239 -240

Line 237 and throughout: p-values, I prefer an actual value rather than <0.05, but I will leave that the editor.

P value stated only if significant, where not significant it is presented as P > 0.05

Line 245:  I disagree that there were relatively few pregnancy complications – just report ;

also what about rate of macrosomia?

Line 263: any association between GWG and age or parity?

leading sentence removed

Although we did not specifically collect data regarding macrosomia, 5% of babies were LGA (> 4.5kg) & this information is now added – see line 264

Thank you we have conducted Chi square test for independence which showed no statistical significance, this has been added into the text, see lines 250 - 252

Table 3 seems to be missing parts of the headers; it was difficult to read; why not include % not meeting LRNI?; %exceeding RNI)?; protein – why no mean pro g at time points but given the RNI.  Also indicate which are EAR and which are RNI; for sugar and FSA, include how many gm/day footnote should include all abbreviations

Table 3 has been revised to reflect reviewer’s comments. RNI and LRNI not applicable to other nutrients in this table. Only protein has an RNI but have amended this to reflect it as a % of total energy based on kcals in line with the other macronutrients.

What happened to Table 4? Check table numbering

Amended

Table 5:Vit D no LRNI?

A footnote re LRNI for vitamin D has been added to (now) Table 4

Discussion

Lines 340-342: please cite “previous literature” and be more specific to outcomes you are talking about

Previous literature is now referred to, see lines 364

Line 346-348: based in your study you cannot make this conclusion.  You had only 39 people with GWG so cannot extrapolate to the population.  Dismissing specific wt guidance is also not the aim of the study since specific guidance was not assessed.  Concur with promoting healthy diet cannot be argued against, but is not defined, so that is not helpful.  Suggest you stick to what you measured with an adequate sample size, since there is info on diet guidance that would be helpful given what you found about macronutrients.

We have reworded this to “Whilst the sample size of GWG findings was underpowered, the results would support the UK’s approach [2] in promoting a healthy diet for pregnant women”. See lines 369 -371

Line 364: Unclear why you could not look at the food records and determine proportion of servings coming from high biological value (you could report pro servings in terms of animal sources if you wanted); regardless since so many met the pro goal, uncertain how much this is relevant at all, so either add info or delete.

We have removed the reference to biological value.

For the whole micronutrient section, need to compare your findings against other studies of obese pregnant women; or compare to obese non-pregnant women if not available in pregnancy.

Data re micronutrients is limited for pregnant women with obesity, plus adding this would make the discussion very long & reviewer 2 has asked for us to shorten it.

We have however referred to WHO (2016), where increasing BMI in pregnancy was associated with multiple micronutrient deficiencies: “This study concurs with WHO guidelines [6] who highlight that the higher the BMI in pregnancy, the greater the risk of micronutrient deficiencies”.

See line numbers 468 - 469

Line337: include consequences of high SFA and added sugars, inadequate fibre in pregnancy

This is added:

“High added sugar and low dietary fibre intakes are associated with a number of non communicable diseases, such as dental caries, obesity and bowel disorders such as constipation, diverticular disease and increased risk of colorectal cancer [50].  See lines 400 - 403

Plus:

“High intakes for SFA are associated with poor cardiometabolic health [51] See line 410

Line 391: compare your findings to other studies in preg women

Reference made to a comparable study. See lines 389 - 390

Line 394: low fe intake is common during pregnancy, hence need for supplementation – need to include information on this guidance; same for Line 421 on folate – include rationale for supplementation worldwide

This has been added:

“Iron deficiency is extremely prevalent globally, particularly among women of childbearing age; hence the use of routine iron supplementation during pregnancy in several countries [6]. Dosage of supplementation varies widely in different countries (ranging from 9-50 mg/day); with WHO (2016) recommending of 27mg/day. However, iron supplementation is not routinely offered to pregnant women in the UK” [52]. See lines 426 - 431

Line 432: look for a relationship between preeclampsia (12%) and LBW (7%) and your variables.  What were the characteristics of those women?  In terms of obesity class, age, dietary intake, GWG?

Chi-square test for independence showed no statistical significant differences, this text has been added (see lines 250 – 252)

Line 462: characteristics Typo?

amended

Line 483: since so few women were referred to a dietitian, add this is an opportunity for midwives to make referrals

We have amended the text

“The RCOG (2018) suggests pregnant women living with obesity to be referred and supported by a dietitian [2], though women with obesity report rarely receiving support [7]. Midwives should  refer pregnant women to a dietician as required. Care pathways need revising so that women living with obesity are supported throughout pregnancy to optimise their nutritional composition, with a view to improving pregnancy and clinical outcomes.” (See lines 509 – 510)

Lines 512-513 – excellent point!

Thank you

Future research should also include role of socio-economic status, ,race,  smoking, education, parity, inflammation, dietary advice/counselling, referral to dietitian, age on dietary intake and GWG and birth outcomes; does micronutrient supplementation mitigate worse birth

We have added these recommendations to future research within the discussion, see lines 544 - 546

Reviewer 2 Report

The authors report the results of a cross-sectional observational study aimed to observe quality of dietary intakes in pregnant obese women (BMI > 35 kg/m2) and to assess the quality of macro and micronutrients intakes.

This subject is very interesting and important in the context of the increase of obesity epidemic. One original point is to focus on obese women because research on quality of diet in such population is limited. They used an original method for the evaluation of quality of food intakes by using recording of 3 days’ diaries at three different time points during pregnancy. They also used an original method to code data by using a food composition database. This allowed to collect a huge amount of data and to produce an overview of the quality of nutrition throughout pregnancy, regarding macro and micronutrients.It is a shame that the GWG could not be analysed, because it was not recorded in such study.

However, the paper is too long, and quite difficult to read. It may be improved by making it shorter and more synthetic. Also, part of the results is not clearly presented.

Introduction is too long and some points developed in this section are repeated in the discussion (line 100-114)

Methods:

There are some issues concerning the population and some information should be added: how were the pregnant women selected? Are they representative of the general population in UK? Did they receive nutritional advice during pregnancy, especially those with GDM? What about women with foreign languages, were they excluded? Is there in the cohort women who underwent bariatric surgery?

Results:

Clinical outcome: Birthweight should be reported as percentile because in this case gestational age at birth is considered. What about prematurity in this cohort? Also, the total percentage of vaginal delivery (57%) and C-section (19%) is not 100%. The value of normal Apgar score should be written in the text (7-10).

Table 3 is not easy to understand, mainly because the title is not appropriate and does not correspond to the results presented. The presentation should be revised, for example the line concerning protein is not understandable. What is P:S ratio? Why Ear appears in few lines of the table?

There is no table 4.

Is it feasible to report the individual variation in quality of food intakes throughout pregnancy, as well as the relation between some specific macro and micro nutrients and birth weight?

Discussion: It is really too long, mainly because it describes most of the results. It should be more synthetic to be easier to read.

Line 363: what is NDNS?

Line 376-378: should we understand that dietary fibre is NSP? The authors refer to recommended level of 30g/day but data about this point are not reported in table 3.

Lines 426-440: This part of the discussion repeats points already mentioned in the introduction. This should appear in one or other section.

Line 414-420: this part could be deleted.

Clinical outcomes: this part could be deleted because it is not the purpose of this study.

Review reference citation line 512.

Line 548: the words “birth weight” are not appropriated because they usually refer to the baby and it is confusing. I suppose the authors mean weight at delivery.

Author Response

Response to reviewer 2:

Comments and Suggestions for Authors

The authors report the results of a cross-sectional observational study aimed to observe quality of dietary intakes in pregnant obese women (BMI > 35 kg/m2) and to assess the quality of macro and micronutrients intakes.

This subject is very interesting and important in the context of the increase of obesity epidemic. One original point is to focus on obese women because research on quality of diet in such population is limited. They used an original method for the evaluation of quality of food intakes by using recording of 3 days’ diaries at three different time points during pregnancy. They also used an original method to code data by using a food composition database. This allowed to collect a huge amount of data and to produce an overview of the quality of nutrition throughout pregnancy, regarding macro and micronutrients.It is a shame that the GWG could not be analysed, because it was not recorded in such study.

Thank you for this feedback

However, the paper is too long, and quite difficult to read. It may be improved by making it shorter and more synthetic. Also, part of the results is not clearly presented.

The submission has been shortened and revised in line with reviewer feedback throughout, this should address this specific comment.

Introduction is too long and some points developed in this section are repeated in the discussion (line 100-114)

The introduction has been shortened and amended, please see sections in red. (see Lines 98 - 102 and 443-449)

Methods:

There are some issues concerning the population and some information should be added: how were the pregnant women selected? Are they representative of the general population in UK? Did they receive nutritional advice during pregnancy, especially those with GDM? What about women with foreign languages, were they excluded? Is there in the cohort women who underwent bariatric surgery?

This section has been rephrased to make it more explicit, particularly about exclusion criteria (see lines 123 -146)

Results:

Clinical outcome: Birthweight should be reported as percentile because in this case gestational age at birth is considered. What about prematurity in this cohort?

Also, the total percentage of vaginal delivery (57%) and C-section (19%) is not 100%. The value of normal Apgar score should be written in the text (7-10).

Thanks for this suggestion – we do not have data on birth weight percentile or gestational age for this cohort. It would be useful to include this in future research.

Thank you for pointing out these typos, the text has been amended. (see table 2 & lines 267)

Table 3 is not easy to understand, mainly because the title is not appropriate and does not correspond to the results presented. The presentation should be revised, for example the line concerning protein is not understandable. What is P:S ratio? Why Ear appears in few lines of the table?

Table 3 has been amended

There is no table 4.

Corrected table numbering throughout 

Is it feasible to report the individual variation in quality of food intakes throughout pregnancy, as well as the relation between some specific macro and micro nutrients and birth weight?

Thank you for this suggestion we have added a comment about the changes of food intakes during pregnancy (see lines 312 -313)

However, we have not conducted regression modelling on the data.  

Discussion: It is really too long, mainly because it describes most of the results. It should be more synthetic to be easier to read.

In line with both reviewers’ comments, the discussion section has been amended. Please see red text

Line 363: what is NDNS?

This has been clarified – it is the National Diet & Nutrition Survey (2016). See line 66)

Line 376-378: should we understand that dietary fibre is NSP? The authors refer to recommended level of 30g/day but data about this point are not reported in table 3.

Data predates change in definition of dietary fibre, but NSP is the major component of dietary fibre. A footnote has been added to table 3 (see line 298)

Lines 426-440: This part of the discussion repeats points already mentioned in the introduction. This should appear in one or other section.

Aspects of the introduction and the discussion have been amended please see text in red.

Line 414-420: this part could be deleted.

 We have deleted this section from the original manuscript.

Clinical outcomes: this part could be deleted because it is not the purpose of this study.

Thank you for this suggestion, however we disagree as believe it adds to understanding of the importance of nutrient intake

Review reference citation line 512.

Format corrected

Line 548: the words “birth weight” are not appropriated because they usually refer to the baby and it is confusing. I suppose the authors mean weight at delivery.

This has been amended, (see line 580), to:

“Self-reported GWG, or the mothers weight at delivery birth,”

Round 2

Reviewer 1 Report

The manuscript has been improved and is acceptable for publication.